# Intracellular Growth Inhibition and Host Immune Modulation of 3-Amino-1,2,4-triazole in Murine Brucellosis

**DOI:** 10.3390/ijms242417352

**Published:** 2023-12-11

**Authors:** Trang Thi Nguyen, Tran Xuan Ngoc Huy, Ched Nicole Turbela Aguilar, Alisha Wehdnesday Bernardo Reyes, Said Abdi Salad, Won-Gi Min, Hu-Jang Lee, Hyun-Jin Kim, John-Hwa Lee, Suk Kim

**Affiliations:** 1Institute of Animal Medicine, College of Veterinary Medicine, Gyeongsang National University, Jinju 52828, Republic of Korea; nguyentrang29071996@gmail.com (T.T.N.); txn.huy@hutech.edu.vn (T.X.N.H.); cntaguilar24@gmail.com (C.N.T.A.); dheerow2323@gmail.com (S.A.S.); wongimin@gnu.ac.kr (W.-G.M.); hujang@gnu.ac.kr (H.-J.L.); hyunjkim@gnu.ac.kr (H.-J.K.); 2Institute of Applied Sciences, HUTECH University, 475A Dien Bien Phu St., Ward 25, Binh Thanh District, Ho Chi Minh City 72300, Vietnam; 3Department of Veterinary Paraclinical Sciences, College of Veterinary Medicine, University of the Philippines Los Baños, Los Baños 4031, Philippines; abreyes4@up.edu.ph; 4College of Veterinary Medicine, Chonbuk National University, Iksan 54596, Republic of Korea; johnhlee@chonbuk.ac.kr

**Keywords:** *Brucella abortus*, 3-AT, catalase inhibitor, intracellular growth, immune response, RAW 264.7 cell, ICR mouse

## Abstract

Catalase, an antioxidant enzyme widely produced in mammalian cells and bacteria, is crucial to mitigating oxidative stress in hostile environments. This function enhances the intracellular survivability of various intracellular growth pathogens, including *Brucella* (*B*.) *abortus*. In this study, to determine whether the suppression of catalase can inhibit the intracellular growth of *B*. *abortus*, we employed 3-amino-1,2,4-triazole (3-AT), a catalase inhibitor, in both RAW 264.7 macrophage cells and an ICR mouse model during *Brucella* infection. The intracellular growth assay indicated that 3-AT exerts growth-inhibitory effects on *B. abortus* within macrophages. Moreover, it contributes to the accumulation of reactive oxygen species and the formation of nitric oxide. Notably, 3-AT diminishes the activation of the nucleus transcription factor (NF-κB) and modulates the cytokine secretion within infected cells. In our mouse model, the administration of 3-AT reduced the *B. abortus* proliferation within the spleens and livers of infected mice. This reduction was accompanied by a diminished immune response to infection, as indicated by the lowered levels of TNF-α, IL-6, and IL-10 and altered CD4^+^/CD8^+^ T-cell ratio. These results suggest the protective and immunomodulatory effects of 3-AT treatment against *Brucella* infection.

## 1. Introduction

Brucellosis is a common zoonotic disease caused by *Brucella* species. Despite its low fatality rate, it imposes a substantial burden in terms of morbidity and economic costs, making it a concern for public health and the economy in regions where it is endemic. Brucellosis is a major concern in livestock, leading to abortion, the death of young ones, and reduced milk production in infected animals [1]. *Brucella*, when present in animals, has the potential to transmit and induce brucellosis in humans, leading to symptoms such as fever, joint pain, and muscle pain. *Brucella* spp. are characterized by their Gram-negative nature and their ability to thrive as facultative intracellular bacteria. This pathogen can evade the host immune system, enabling it to survive and replicate within host cells. It can multiply within both professional phagocytes and non-professional phagocytes. However, professional phagocytes, such as dendritic cells, macrophages, and neutrophils, are more efficient at eliminating the ingested *Brucella* [2].

The neutrophil function is the initial defense against *Brucella*, while macrophages come into play as the secondary defense and can act as a reservoir for *Brucella* infection [3]. After *Brucella* entry, macrophages produce an array of microbial compounds and free radicals, including reactive oxygen species (ROS), superoxide anions (O_2_^•−^, hydrogen peroxide (H_2_O_2_), and hydroxyl (OH), to combat the invading bacteria [4]. In order to survive in hostile intracellular environments, *Brucella* expresses Cu-Zn SOD and catalase under oxidative stress inside the host cell [5,6]. Notably, periplasmic catalase plays a critical role in protecting *Brucella* from H_2_O_2_ in specific conditions within the host cells [7]. Furthermore, macrophages themselves rely on catalase to regulate oxidative compounds and safeguard themselves against oxidative stress and damage [8]. Thus, catalase is essential for both *Brucella* and macrophages during infection.

Catalase is a common antioxidant enzyme with the highest turnover rate among all aerobic organisms. This heme enzyme converts H_2_O_2_ into water and oxygen, thereby reducing the toxic effects of H_2_O_2_ and protecting cells from excessive damage [9]. However, catalase in the periplasm of *Brucella* also plays a crucial role in protecting the bacteria from H_2_O_2_ produced during the host cell’s response to brucellosis. The supplementation of catalase to *Brucella*-infected macrophages enables the survival of *Brucella* [4]. Therefore, controlling the catalase activity may contribute to the management of brucellosis.

The compound 3-amino-1,2,4-triazole (3-AT) is known as a specific catalase inhibitor that suppresses catalase activity by covalently binding to the active center of the tetradic form [10]. It has been reported that 3-AT inhibits the peroxisomal transportation of catalase in human skin fibroblast cell lines [11]. Ruiz-Ojeda et al. have reported that 3-AT induces the production of ROS and elicits an inflammatory response in human adipocytes [12], which is closely related to the host’s response to *Brucella* infection. Moreover, 3-AT has exhibited beneficial effects, including anti-obesity effects, improvements in the metabolic status in mice [13], and the inhibition of *Escherichia coli* growth [14]. Although prior studies have shed light on the potential effects of 3-AT and its role in catalase inhibition concerning the modulation of the immune response and bactericidal activity, none have explored its effects on bacterial infectious diseases. Thus, we conducted an investigation utilizing RAW 264.7 cells and an ICR mice model to determine whether 3-AT can enhance the host protection against *Brucella* (*B*.) *abortus* infection or diminish the persistence of this pathogen within the host.

## 2. Results

### 2.1. 3-AT Treatment Inhibits the Intracellular Growth of B. abortus in RAW 264.7 Cells

To investigate the effect of catalase inhibitors on the invasion and survivability of *Brucella* within macrophages, we performed internalization and intracellular growth assays using 3-AT on infected RAW 264.7 cells. The two highest concentrations of 3-AT that did not induce cytotoxicity (3 and 6 mM) (Figure 1A) were applied for the assays, using PBS as a control. The numbers of *B. abortus* phagocytosed by RAW 264.7 cells pre-treated with 3 mM and 6 mM of 3-AT and PBS in the internalization assays were similar, indicating that the 3-AT treatment did not affect the invasion of *Brucella* into macrophages (Figure 1C), whereas, at the same concentrations, 3-AT significantly reduced the intracellular growth of *B. abortus* by 1.45- and 1.88- log10 at 48 h post-infection (pi), respectively, compared to control cells (Figure 1D). Notably, these concentrations did not exhibit a bactericidal effect on *B. abortus* (Figure 1B). These findings reveal the inhibitory effect of 3-AT on the intracellular growth of *B. abortus* in RAW 264.7 cells without the induction of a bactericidal effect.

### 2.2. 3-AT Treatment Affects ROS Accumulation and NO Production in RAW 264.7 Cells

We conducted a catalase activity assay to confirm the catalase inhibitory effect of 3-AT in RAW 264.7 cells and *B. abortus*. In the RAW 264.7 cells, the catalase activity was slightly lower in 3-AT-treated cells compared to PBS-treated cells at 48 h pi (Figure 2A). The presence of 3-AT in *Brucella* culture broth reduced the bacterial catalase activity by 50% compared to the control (Figure 2B). Because the reduction in catalase activity can lead to ROS accumulation [15], which plays an essential role in combating *Brucella*, an ROS detection assay was performed using the fluorescence signal derived from DCFH-DA to determine whether 3-AT triggers ROS accumulation. We found that 3-AT increased the fluorescence intensity in infected cells by 2.4-fold compared to the control (Figure 2C), indicating an elevation in the ROS levels in macrophages. To further investigate the effect of 3-AT on another crucial *Brucella*-killing factor, namely, NO, we used the Griess reagent system to perform NO assays. Interestingly, treatment with 3-AT reduced the NO production in *B. abortus*-infected RAW 264.7 cells (Figure 2D). These findings suggest that 3-AT primarily inhibits the intracellular growth of *B. abortus* within macrophages by inducing ROS accumulation, without significantly affecting the NO production.

### 2.3. Catalase Inhibition by 3-AT Prevents B. abortus from Inducing NF-κB Activation and Modulates Cytokine Secretion in RAW 264.7 Cells

It has been shown that ROS can both stimulate and suppress the activation of NF-κB, which plays a central role in the host response to pathogenic infections [16]. Therefore, we performed an immunoblotting assay to investigate the effect of 3-AT on the activation of NF-κB. Figure 3 shows that the expressions of NF-κB p50 and p65 in the nuclei of *B. abortus*-infected cells at 48 h pi were significantly reduced in the presence of 3-AT. Considerable evidence suggests that NF-κB plays a critical role in the regulation of many cytokines during infections. Thus, we performed qRT-PCR to assess the mRNA expressions of some essential cytokines in *Brucella* infections. We found that 3-AT decreased the expressions of TNF-α and IL-10 (Figure 4A,D) but did not affect the expression levels of IL-6 or IL-12 p40 (Figure 4B,C) in *B. abortus-*infected RAW 264.7 cells. However, serum cytokine measurements showed that 3-AT treatment did not significantly affect TNF-α (Figure 4E) or IL-12 (Figure 4G) production but reduced the production of IL-6 (Figure 4F) and IL-10 (Figure 4H) in infected macrophages at 48 h pi. These results suggest that 3-AT inhibits catalase activity, leading to reduced NF-κB activation and interference with the expressions of key cytokines in the control of brucellosis.

### 2.4. 3-AT Treatment Enhances Protection in Mice against B. abortus Infection and Reduces Excessive Inflammation

The results of the in vitro experiments revealed the potential effect of 3-AT against *Brucella* infection. Hence, we applied 3-AT to a mouse model to investigate its potential in controlling murine brucellosis. The 3-AT was pre-administered and intermittently given to *B. abortus*-infected ICR mice at 400 mg/kg/day or 600 mg/kg/day for 12 days before they were sacrificed, using PBS as the vehicle control. A significant reduction in the numbers of *B. abortus* were observed in the spleens and livers of 3-AT-treated mice. The 3-AT treatment reduced the bacterial loads by 1.2log10–1.5log10 in the spleens and by 1.4log10–1.7log10 in the livers of *Brucella*-infected mice in comparison to the control. The total weights of the spleens and livers collected from mice administered 3-AT were significantly reduced compared to those of the control group (4.3-fold lower in the spleens and approximately 1.54-fold lower in the livers) (Figure 5B), suggesting that 3-AT treatment inhibits the proliferation of *B. abortus* in infected mice. The CD4^+^/CD8^+^ ratio reflects the functional status of effector T cells, which are vital for the adaptive immune response against *Brucella* infection. In this study, the CD4^+^/CD8^+^ ratio in each mouse group was measured (Figure 5C). The CD4^+^/CD8^+^ ratio was higher in the non-infected groups (5.2 ± 0.5, 5.98 ± 1.82, and 6.01 ± 2.25 in the PBS, 3-AT at 400 mg/kg/day, and 3-AT at 600 mg/kg/day groups, respectively) compared to the infected group. Additionally, the ratio was higher in the group treated with 3-AT at a dose of 600 mg/kg/day (4.72 ± 0.39) compared to the control group (3.79 ± 0.68).

In addition to T-cell differentiation, the cytokine levels in the blood reflect the status of the host immune responses and disease progression upon *Brucella* infection [17]. This study quantified six cytokines affected during *Brucella* infection in the cardiac blood using CBA kits and flow cytometry: TNF-α; IFN-γ; IL-6; IL-12p70; IL-10; and MCP-1. As shown in Figure 6, the TNF-α, IFN-γ, IL-6, and IL-10 levels in the mice administered 3-AT were significantly lower than those in the control group. In contrast, the levels of IL-12p70 and MCP-1 in the 3-AT-administered and control groups were not significantly different. These results suggest the protective and immunomodulatory effects of 3-AT on *B. abortus* infection in mice.

## 3. Discussion

It is well established that *B. abortus* catalase protects against hydrogen peroxide in the host intracellular environment, which is one of the influential host-killing factors in response to *Brucella* infection [7]. Moreover, the supplementation of catalase has been indicated to increase the intracellular growth of *B. abortus* within macrophages [4]. These findings have led to new approaches aimed at arresting the survival and replication of intracellular *Brucella*. Our research findings suggest that the inhibition of catalase by 3-AT enhances the host defense in murine brucellosis. We found that 3-AT treatment can inhibit the intracellular growth of *B. abortus* within macrophages. This may be attributed to the catalase inhibitory action of 3-AT, resulting in the increased accumulation of cellular ROS and the heightened H_2_O_2_ sensitivity of *B. abortus.* Notably, 3-AT administration reduced the numbers of *B. abortus* in the spleens and livers, the most visibly infected organs in the mice. The reduction in the levels of some inflammation cytokines, such as TNF-α, IFN-γ, and IL-6, as well as the increase in the CD4^+^/CD8^+^ ratio in the peripheral blood of 3-AT-treated mice during *Brucella* infection, illustrate the immunomodulatory effect of 3-AT.

It has been reported that 3-AT treatment, in the presence of the TLR-2 antibody, inhibits the intracellular growth of *Staphylococcus aureus* within murine peritoneal macrophages [18]. Interestingly, we observed that only the 3-AT treatment significantly reduced the intracellular *B. abortus* survival by up to 98% within the macrophage RAW 264.7 cells. However, it did not exhibit a bactericidal effect on this pathogen, suggesting that 3-AT treatment enhances macrophage defense mechanisms against *Brucella* or reduces its adaptive capabilities within host cells. Furthermore, the inhibition of the catalase activity of intracellular *B. abortus* possibly increases the sensitivity of this bacterium to the H_2_O_2_ present in the host cell’s intracellular environment, produced by the host cell, as reported in a study conducted by Kim et al. [7].

Previous reports have indicated that 3-AT augments the intracellular ROS or hydroperoxide of various cell types, such as human differentiated adipocytes, macrophage polarization in adipose tissue, lung cancer Calu-6 cells, and fibroblast cells [12,19,20,21]. These reports are consistent with our findings. The treatment of 3-AT on RAW 264.7 cells led to intracellular ROS accumulation, although the catalase activity of the RAW 264.7 cells did not significantly decrease under the 3-AT treatment. In contrast, 3-AT treatment significantly decreased the *Brucella* catalase activity, suggesting that the catalase inhibitory effect of 3-AT is more effective against *Brucella*.

ROS are recognized as a vital killing factor in the innate immune response against *Brucella* infection in immune cells [4]. Additionally, the reduction in the catalase activity in *Brucella* can contribute to inhibiting its replication within the intracellular environment, as described of *Helicobacter pylori* in macrophages [22]. The dual impact of high intracellular ROS in RAW 264.7 cells and the reduction in catalase inside the bacteria lead to increased oxidative sensitivity and significantly diminish the *Brucella* replication within macrophages.

Apart from ROS, NO also plays an essential role in the clearance of intracellular *Brucella* [4]. However, unlike ROS, the level of NO released into the culture supernatant of *B. abortus*-infected cells was reduced in the presence of 3-AT, indicating that 3-AT inhibits NO production. This result is in line with the results described by Buchmuller and colleagues [23], who demonstrated that 3-AT inhibited NO synthesis. Although 3-AT reduced the NO production, the intracellular *B. abortus* significantly decreased. This may be because NO’s lethal effect is not all-encompassing [24], and the elimination of NO by *Brucella* is most effective during the initial 24 h of infection. Subsequently, surviving bacteria may employ genetic mechanisms to deal with the enriched NO environment or utilize NO as a nitrogen source [25]. In addition, NO reduction in infected RAW 264.7 cells may contribute to protecting cells from damage caused by the increasing intracellular ROS [26].

NF-κB, a family of transcription factors comprising RelA (p65), NF-κB1 (p50 and p105), NF-κB2 (p52 and p100), c-Rel, and RelB, plays a central role in a variety of cellular regulatory processes. The activation status of NF-κB depends on its interaction with its inhibitor, IκB. In normal conditions, NF-κB remains inactive. However, when stimulated, NF-κB is liberated from its inhibitor and migrates into the nucleus to initiate various functions, including the regulation of cytokines, such as IL-6, IL-10, and TNF-α [27]. The activation of NF-κB is crucial for mounting an effective immune response against *Brucella* invasion [28,29].

We found that the 3-AT treatment suppressed the translocation of NF-κB p65 and p50 into the nuclei of *B. abortus*-infected cells at 48 h pi. Similarly, a study conducted by Mu and associates [30] indicated that the suppression of NF-κB activation by 3-AT leads to the impediment of the mRNA expressions of pro-inflammatory cytokines, possibly due to ROS accumulation. Note that the accumulation of ROS can activate or repress NF-κB through different pathways [31]. ROS can directly regulate the NF-κB heterodimer, oxidize NF-κB, and subsequently diminish its DNA-binding ability. The cysteine residue (Cys-62) in the domain of NF-κB p50 is sensitive to oxidation, leading to a reduction in the DNA-binding ability of p50 and its activation under conditions of increased intracellular ROS [32,33,34]. In the case of p65, the phosphorylation of Ser-276 of this subunit is easily affected by ROS. Furthermore, ROS obstruct the phosphorylation of IκBα, preventing its degradation and the subsequent translocation of NF-κB into the nucleus [31]. NF-κB p105 and p100 are precursors of NF-κB p50 and p52, respectively, and are primarily localized in the cytosol in their unprocessed forms, regulated by canonical pathways [31]. Interestingly, Figure 3 shows the strong expressions of NF-κB p105 and p100 in the nuclei of *B. abortus*-infected cells, possibly due to the cellular stress triggered by the infection at 48 h pi. The cellular stress and infection can activate alternative NF-κB regulation, cytokine secretion, ROS accumulation, and other signaling pathways, leading to the presence of NF-κB p105 and p100 in the nucleus.

The exposure to 3-AT results in an elevation of the intracellular ROS levels and hinders the nucleus translocation of NF-κB p50 and p65. Consequently, this alteration affects the mRNA expressions of the genes regulated by these NF-κB subunits. As an example, IL-10, an anti-inflammatory cytokine regulated by NF-κB p50 [35], showed a significant reduction in both the mRNA level and protein expression in the *B. abortus*-infected macrophages following 3-AT treatment in the present study. This reduction is associated with the decrease in the presence of NF-κB p50 protein within the nucleus. In *B. abortus* infection, IL-10 suppresses lysosome-mediated bacterial clearance, thereby enhancing the survivability of the pathogen within macrophages [36]. The reduction in the IL-10 levels in RAW 264.7 cells under 3-AT treatment may inhibit *B. abortus* replication within host cells. The inhibition of NF-κB p65 activation has varying effects on the expressions of other key cytokines involved in controlling *Brucella* infection, such as TNF-α, IL-6, and IL-12 [37,38,39].

The mRNA level of TNF-α was decreased, consistent with previous studies [30], while the cytokine level remained relatively unchanged in the presence of 3-AT. In contrast, the gene expression of IL-6 did not significantly change, but its protein expression was decreased. The 3-AT treatment did not affect IL12-p70 in either the mRNA or protein levels. The inverse correlation between the mRNA and protein levels of TNF-α and IL-6 has been reported in ovarian cancer cells by Israelsson and colleagues [40], which is possibly due to the complex regulatory mechanisms governing gene transcription, protein translation, and the stability of these cytokines. Additionally, unknown negative-feedback mechanisms may be in place to regulate mRNA and protein expression [40]. Pro-inflammatory cytokines are known to contribute to the clearance of *Brucella* in macrophages. However, based on our results, the IL-6 levels do not coincide with a decrease in the intracellular growth of *B. abortus* in the context of 3-AT treatment, suggesting that IL-10 reduction may have a more significant impact on intracellular *B. abortus* growth compared to IL-6 levels.

The pleiotropic effects of 3-AT in the context of *Brucella* infection on RAW 264.7 cells, which include the inhibition of intracellular growth bacteria, increased the ROS accumulation, and the modulation of the cell’s immune response suggests 3-AT as a potential treatment to control brucellosis. Consequently, 3-AT was administered to ICR mice to investigate its protective effects against murine brucellosis. Intermittent treatment with 3-AT led to a reduction in the bacterial loads in the spleens and livers of mice challenged with *B. abortus*. This study first described the effectiveness of 3-AT at reducing the pathogen proliferation within host organs. The cytokine levels and the CD4^+^/CD8^+^ ratio pattern can serve as indicators of the immune status in brucellosis. In the acute phase of brucellosis, infected patients exhibit higher levels of cytokines, such as TNF-α, IL-6 or IFN-γ, and IL-10, compared to *Brucella*-negative individuals [41,42,43,44]. In the convalescent stage, these cytokines are reduced compared to pre-treated patients but remain elevated compared to healthy cases [43]. Our study revealed that the serum concentrations of TNF-α and IL-6 or IFN-γ and IL-10 in *B. abortus*-infected mice receiving 3-AT were notably reduced compared to those of the infected group without 3-AT treatment, yet they were still higher than the levels observed in the non-infected group. These results are consistent with the cytokine profiles observed in human brucellosis [41,42,43].

We also observed a similar CD4^+^/C8^+^ ratio pattern in the 3-AT-treated group in both *Brucella*-infected and non-infected mice, mirroring the patterns seen in patients receiving pre- and post-antibiotic treatment during acute brucellosis and in healthy individuals. This correlation reflects the recuperation of the T-cell function following the dysfunction caused by *Brucella* infection [45,46,47]. Although in vivo results show the protective effects of 3-AT against *B. abortus* infection, the precise mechanism underlying this protective ability and how it modulates the immune response in infected mice remain unclear. Hence, further experiments should be conducted to address these unanswered questions.

## 4. Materials and Methods

### 4.1. Reagents

The 3-AT (A8056), MTT (M5655), triton ×100 (9002-93-1), 30% hydrogen peroxide (H1009), and catalase assay kit (MAK-381-1KT) were purchased from Sigma-Aldrich (Seoul, Republic of Korea). DCFA/H2DCFDA-cellular ROS assay kit (ab113581-Abcam, Cambridge, UK), Griess reagent system (G2930), and GoTaq^®^qPCR master mix (A6002) were purchased from Promega (Fitchburg, WI, USA). NP-40 (85124) and cell extraction buffer (FNN0011) were acquired from ThermoFisher Scientific (Branchburg, NJ, USA). Cocktail (P3200-005) and bovine serum albumin (A0100-010) were purchased from GenDEPOT (Barker, TX, USA) and EzWestLumi plus (WSE-7120L, Atto, Tokyo, Japan). RiboEx (301-001) was purchased from Geneall (Gyeonggi, Republic of Korea). QuantiTect^®^ Reverse transcription kit (205311) was purchased from (Qiagen, Hilden, Germany). CBA mouse inflammation kit (552364), CD8a monoclonal antibody (553032), and CD4 monoclonal antibody (17504042) were purchased from Biosciences (San Diego, CA, USA). Red blood cell lysis buffer (11814389001) was purchased from Roche (Basel, Switzerland). All primary antibodies, including NF-κB p65 (8242T), lamin B1 (13435S), NF-κB p105/p50 (12540S), and NF-κB p100/52 (37359S), are rabbit antibodies and were purchased from Cell Signaling (Danvers, MA, USA). Anti-rabbit IgG HRP-conjugated secondary antibody (7074S) was purchased from Cell Signaling.

### 4.2. Cell Culture and Bacterial Growth Conditions

RAW 264.7 cells (ATCC, TIB-71) were grown at 37 °C in a 5% CO_2_ atmosphere in RPMI 1640 medium (1600-044, Gibco, Chino, CA, USA) in the presence of 10% (vol/vol) fetal bovine serum (FBS) (1600-044, Gibco). The smooth, virulent, wild type of *B. abortus* was inoculated in *Brucella* broth (BBL BD, San Jose, CA, USA) at 37 °C in a shaking incubator until it reached the stationary phase before proceeding to the cell infection or mouse infection.

### 4.3. Cell Viability Assessment Assay

An MTT assay was performed to determine the non-cytotoxic concentration of 3-AT. RAW 264.7 cells were seeded at a density of 3 × 10^4^ cells per well in a 96-well plate for 24 h. After that, different concentrations of 3-AT (1, 2, 3, 4, 6, 8, 10, 20, 40, and 60 mM) were added to the cell culture medium for 48 h. After 48 h, the medium was replaced with a new medium containing 5 mg/mL of MTT solution. Following a 4 h incubation, the medium was removed and replaced with 150 µL of DMSO, which was incubated for 15 min. A plate reader (Thermo Labsystems Multiskan, ThermoFisher Scientific, Daejeon, Republic of Korea) was used to measure the absorbance at a wavelength of 540 nm.

### 4.4. Bactericidal Assay

An amount of 1 × 10^4^ colony-forming units (CFUs)/mL of *B. abortus* contained in a 96-well plate was treated using different concentrations of 3-AT (3, 6, and 12 mM) at 37 °C for 0, 4, 8, 24, and 48 h. PBS was used as the vehicle control. At indicated time points, *B. abortus* was diluted 100 times and plated on an agar plate. The number of *B. abortus* colonies was counted after 72 h incubation. The bactericidal effect of 3-AT on *Brucella* was expressed as the percentage of surviving bacteria in the 3-AT treatment compared to the vehicle control.

### 4.5. Internalization and Intracellular Growth Assays

RAW 264.7 cells were cultured at a density of 3 × 10^4^ cells per well for 24 h. PBS or 3-AT was administered to the cells 2 h prior to infection with *B. abortus*, with a multiplicity of infection (MOI) of 50 for the internalization assay. Cells were then washed twice with PBS at 5, 25, and 60 min pi and incubated in a new medium containing 50 μg/mL of gentamicin for 30 min to eliminate extracellular bacteria. The cells were then washed twice using PBS and lysed in distilled water (DW) to release phagocytosed bacteria. The lysates were spread on the *Brucella* agar plate to check bacterial CFUs. Bacterial CFUs were then converted to log10.

The intracellular assay was performed in the same way as the invasion assay, with some modifications. Briefly, RAW 264.7 cells were cultured in the fresh media containing 50 μg/mL of gentamicin in the presence of 3-AT only at 1 h pi. The numbers of intracellular *B. abortus* were determined at 4, 24, and 48 h pi.

### 4.6. Catalase Activity Assay

The catalase activity measurement in *B. abortus* was performed as previously described [48]. Briefly, the *B. abortus* was shaken and inoculated in 5 mL of *Brucella* broth supplemented with 3 mM of 3-AT for 24 and 48 h at 37 °C. At the indicated time, 1 mL of *B. abortus* was collected, washed twice, and suspended in 100 μL PBS. A mixture including 100 μL of triton ×100 and 100 μL of 30% hydrogen peroxide was mixed thoroughly with a *B. abortus* suspension in a Pyrex tube (13 mm diameter, 100 mm height, borosilicate glass; Corning, NY, USA) and then incubated at room temperature. The height of the O_2_-forming foam after remaining steady for 10 min was measured using a ruler. The catalase activity was qualified based on the standard curve with the defined activity unit.

Catalase activity in RAW 264.7 cells was determined using the catalase assay kit, according to the manufacturer’s instructions. Briefly, 10^6^ cells were cultured for 24 h and then infected with *B. abortus* for 1 h, followed by 3-AT (3 mM) treatment. After 24 and 48 h treatment, cells were washed twice using PBS before being lysed in assay buffer. The lysate was centrifuged at 10,000× *g* for 15 min at 4 °C to collect supernatant. The collected supernatant was acquired for catalase activity measurement.

### 4.7. ROS and NO Detection Assay

In the ROS assay, cells were subcultured at a density of 2.5 × 10^4^ cells per well in a 96-well optical-bottom plate for 24 h before infection, followed by 3-AT (3 mM) treatment for 48 h. Following the guidance of the company, the cellular ROS assay was determined using the DCFA/H2DCFDA-cellular ROS assay kit. The fluorescent signal was detected using an F-4500 Fluorescence Spectrophotometer. An NO detection assay was performed to assess the NO concentration produced by RAW 264.7 cells in cell culture media. The assay was conducted using the Griess reagent system following the manufacturer’s instructions.

### 4.8. Extraction of Nucleus Protein

RAW 264.7 cells were washed twice using PBS following 24 and 48 h infection and treatment with 3-AT. Cells were then collected in 500 μL of 1× hypotonic buffer (20 mM tris-HCl; 10 mM NaCl; 3 mM MgCl_2_) and placed on ice for 15 min prior to the addition of 25 μL of 10% NP-40, and were then homogenized via vortex at maximum speed. The homogenate was centrifuged at 3000 rpm for 10 min at 4 °C to obtain the pellet. A complete cell extraction buffer supplemented with protease inhibitor cocktail and PMFS was added to the pellet and incubated on ice with interval vortex every 10 min, repeated three times. Finally, nucleus protein was obtained after centrifuging for 30 min at 14,000× *g* at 4 °C. Protein was qualified using the BCA protein quantification method and stored at −70 °C for the subsequent experiments.

### 4.9. Immunoblotting Assay

Nucleus protein was subjected to SDS-PAGE and transferred onto an immobilon-P membrane (Millipore, Burlington, MA, USA). The membrane was blocked for 30 min, using a blocking buffer containing 5% bovine serum albumin (BSA) suspended in Tris-buffer saline containing 1% of Tween-20 (TBS-T buffer). After that, we incubated the membrane in diluted primary antibodies with a ratio of 1:1000 overnight at 4 °C. All primary antibodies were rabbit anti-mouse antibodies. Following incubation with the primary antibodies and washing three times, the membrane was incubated with a goat anti-rabbit IgG HRP-conjugated secondary antibody with a 1:2000 ratio for 1 h at room temperature. EzWestLumi plus was used to detect the signal, and a Molecular Imager^®^ ChemiDoc^TM^XRS+ system machine (Bio-Rad Laboratories, Carlsbad, CA, USA) was used to visualize the protein expression.

### 4.10. RNA Extraction and Quantitative Real-Time PCR

RAW 264.7 cells were collected in 1 mL of cold RiboEx and homogenized via vigorous vortexing after adding 400 μL of chloroform. The homogenate was centrifuged at 12,000 rpm at 4 °C for 15 min. After centrifugation, the transparent top layer containing RNA was collected to mix with 400 μL isopropanol for 10 min at room temperature. The mixture was transferred into a binding column and centrifuged at 12,000 rpm at 4 °C for 30 min. The RNA bound to the column was washed twice with an RPE buffer and eluted with nuclease-free water. cDNA synthesis was performed using a QuantiTect^®^ Reverse Transcription kit and following the guidance instructions. The real-time PCR was performed using the GoTaq^®^qPCR master mix on a CFX Opus 96 real-time PCR system. The total 20 μL reaction consisted of 2 μL of cDNA, 2 μL each of forward and reverse primers, 10 μL of qPCR master, and 4 μL of DW. The primer sequences are listed in Table 1. The relative levels of mRNAs were analyzed using Bio-Rad CFX Maestro 2.2 software (Version: 5.2.008.0222, Biao-Rad Laboratories, Hercules, CA, USA).

### 4.11. Infection of Mice with B. abortus and Mice Treatment with 3-AT

Seven-week-old female ICR mice were purchased from Samtako (Osan, Republic of Korea). The animal experiment protocol in this study was approved by the Animal Ethical Committee of Chonbuk National University (authorization number: CBNU-2021-037). After one week of acclimation, mice were administered 3-AT via oral gavage at a dose of 400 mg/kg/day or 600 mg/kg/day three days before infection. PBS was used as a control. For infection, mice were intraperitoneally injected with 2 × 10^5^ CFUs of *B. abortus* or PBS. Mice were treated with 3-AT or PBS according to a schedule of two consecutive days of treatment, followed by a one-day rest, for 12 days pi. Mice were sacrificed on the 12th day pi for heart blood, spleen, and liver collection. Collected organs were homogenized in 1 mL of PBS to examine the *B. abortus* growth in the mouse tissues within these organs. Serial dilution of homogenates was prepared in PBS and plated on the *Brucella* agar plate. Colonies were enumerated after 72 h incubation at 37 °C. The number of CFUs was subjected to log10 base. The collected blood was subjected to serum cytokine measurement and CD4^+^/CD8^+^ T-cell population differentiation experiment 4.12 with serum cytokine measurement.

The serum was obtained by centrifuging blood at 2000 rpm at 4 °C for 10 min. The concentrations of cytokines, including TNF-α, INF-γ, IL-6, IL-12p70, IL-10, and MCP-1, were measured using a CBA mouse inflammation kit, following the manual instruction. Data were acquired and analyzed using a BD FACVerse flow cytometer and FCAP array software (Version 3.0, BD Biosciences, Franklin Lakes, NJ, USA).

### 4.12. CD4+/CD8+ T-Cell Ratio in Peripheral Blood

The ratio of differential CD4^+^/CD8^+^ T cells was defined in 100 μL of blood. Blood was incubated with a 75 μL mixture of PE-conjugated rat anti-mouse CD8a monoclonal antibodies and CD4 monoclonal antibodies in 1% BSA for 30 min in the dark at RT. The red blood in the mixture was lysed by adding 2 mL of red blood cell lysis buffer and incubated for 10 min. Three milliliters of PBS was added to terminate the lysis reaction. White blood cells were then collected by centrifuging at 380× *g* for 5 min, washed twice with 3 mL of PBS, and suspended in 0.5 mL PBS. White blood cells were analyzed using a BD FACVerse flow cytometer to determine the populations of CD4^+^ and CD8^+^ T cells, and the CD4^+^/CD8^+^ T-cell ratio was analyzed using BD FACSuite software (version 1.2.1, BD Biosciences, Franklin Lakes, NJ, USA)

### 4.13. Statistical Analysis

The data are presented as the mean ± standard deviation. Statistical analysis was conducted using an unpaired Student’s *t*-test through GraphPad Instat (version 3.00, GrapPad Software, Boston, MA, USA). Significance levels were denoted as follows: * *p* < 0.05, ** *p* < 0.01, and *** *p* < 0.001, indicating the statistically significant differences between the groups.

## Figures and Tables

**Figure 1 ijms-24-17352-f001:**
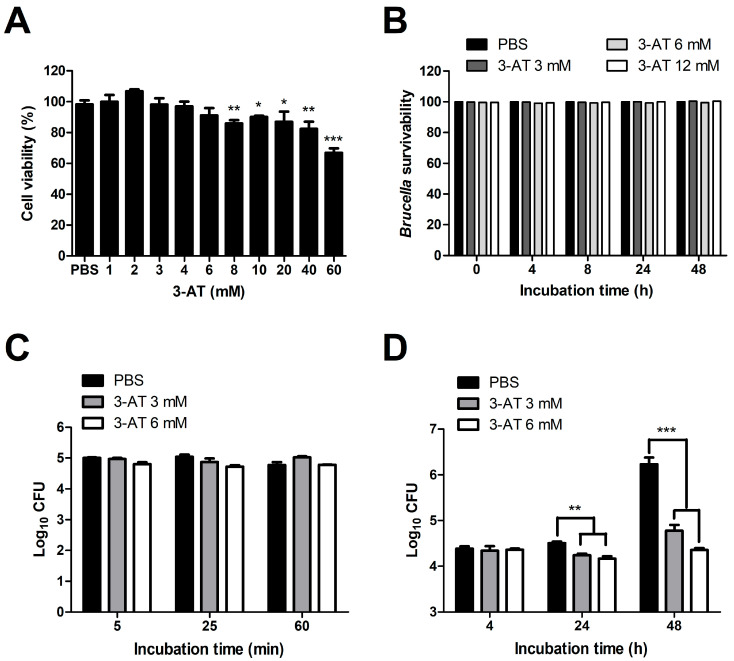
Effect of 3-AT on the internalization and intracellular growth of *B. abortus* within RAW 264.7 cells. Cytotoxic effects of 3-AT on RAW 264.7 cells at 48 h after treatment were assessed via MTT assay (**A**). The bactericidal effect of 3-AT on *B. abortus* was evaluated at different concentrations and incubation time points (0, 4, 8, 24, and 48 h) (**B**). The internalization (**C**) and intracellular growth (**D**) of *B. abortus* within macrophages were evaluated at various time points after pre-treatment with 3 mM and 6 mM of 3-AT. The data are represented as the mean ± SD of replicated samples obtained from at least two independent experiments. Significant deviations from the control group are denoted by asterisks (* *p* < 0.05, ** *p* < 0.01, *** *p* < 0.001).

**Figure 2 ijms-24-17352-f002:**
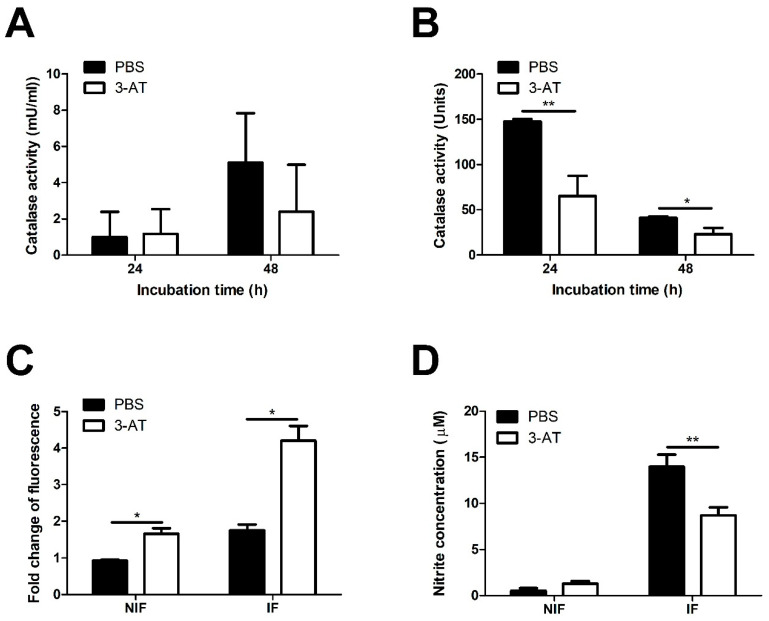
Catalase inhibition affects ROS accumulation and NO production of RAW 264.7 cells. Catalase activities of cells (**A**) and *Brucella* (**B**) were measured after 24 and 48 h exposure to 3 mM of 3-AT or PBS. The effect of the 3-AT treatment on the ROS accumulation in macrophages at 48 h pi was quantified using a DCFA/H2DCFDA-cellular ROS assay kit and spectrofluorometer (**C**). Nitrite production was quantified using the Griess assay and spectrophotometry (**D**). Data are displayed as the mean ± SD of triplicate samples from at least two independent experiments. Statistically significant differences compared to the control group are indicated by asterisks (* *p* < 0.05, ** *p* < 0.01).

**Figure 3 ijms-24-17352-f003:**
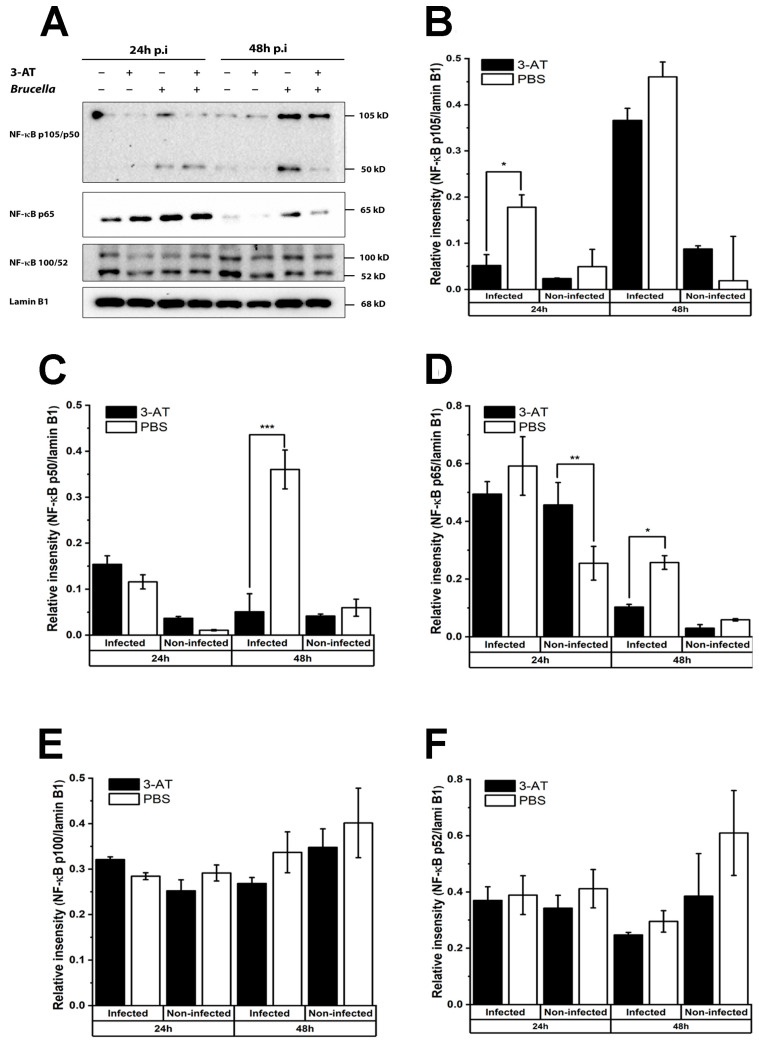
3-AT inhibits the translocation of NF-κB into the nucleus of RAW 264.7 cells. Total protein extracted from nuclear macrophages after 24 and 48 h exposure with 3 mM OF 3-AT or PBS was subjected to the immunoblotting assay (**A**) for detecting the nuclear translocation of NF-κB p105/50, p65, and p100/52. The relative intensities of NF-κB p105/50 (**B**,**C**), p65 (**D**), and p100/52 (**E**,**F**) proteins were normalized to the control lamin B1 using ImageJ 1.5r software, Wayne Rasband (National Institutes of Health, USA). The data are presented as the mean ± SD of duplicate samples obtained from two independent experiments. Asterisks indicate significant deviations (* *p* < 0.05, ** *p* < 0.01, *** *p* < 0.001).

**Figure 4 ijms-24-17352-f004:**
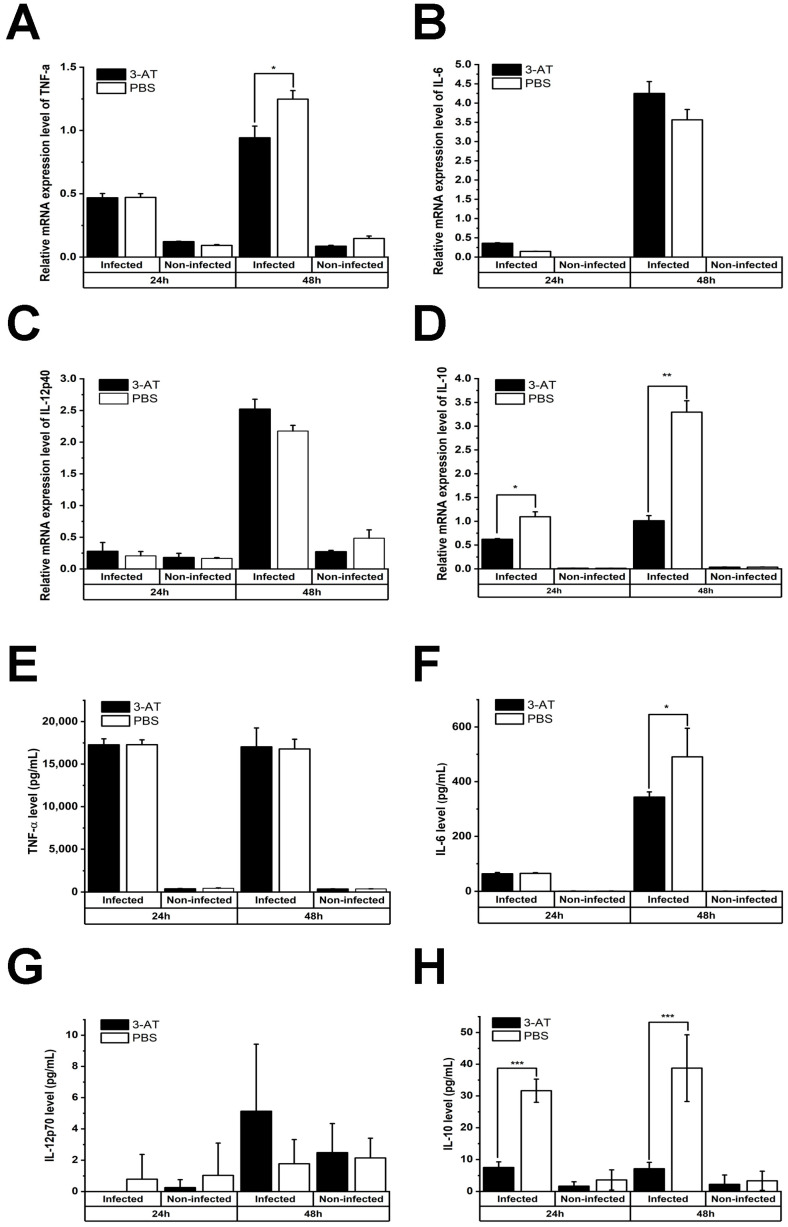
The compound 3-AT induces mRNA expression in macrophages infected with *B. abortus*. Cells were treated with 3-AT at 3 mM for 24 and 48 h. Following treatment, RNA was isolated from RAW 264.7 cells at 24 and 48 h incubation with 3-AT and converted to cDNA. TNF-α (**A**), IL-6 (**B**), IL-12p40 (**C**), and IL-10 (**D**) gene expressions were then evaluated via qRT-PCR. The TNF-α (**E**), IL-6 (**F**), IL-12p40 (**G**), and IL-10 (**H**) cytokine levels in the cell culture media were quantified using a mouse inflammation kit and flow cytometry. Data present the mean ± SD of replicate samples from three independent experiments. Asterisks indicate significant deviations (* *p* < 0.05, ** *p* < 0.01, *** *p* < 0.001).

**Figure 5 ijms-24-17352-f005:**
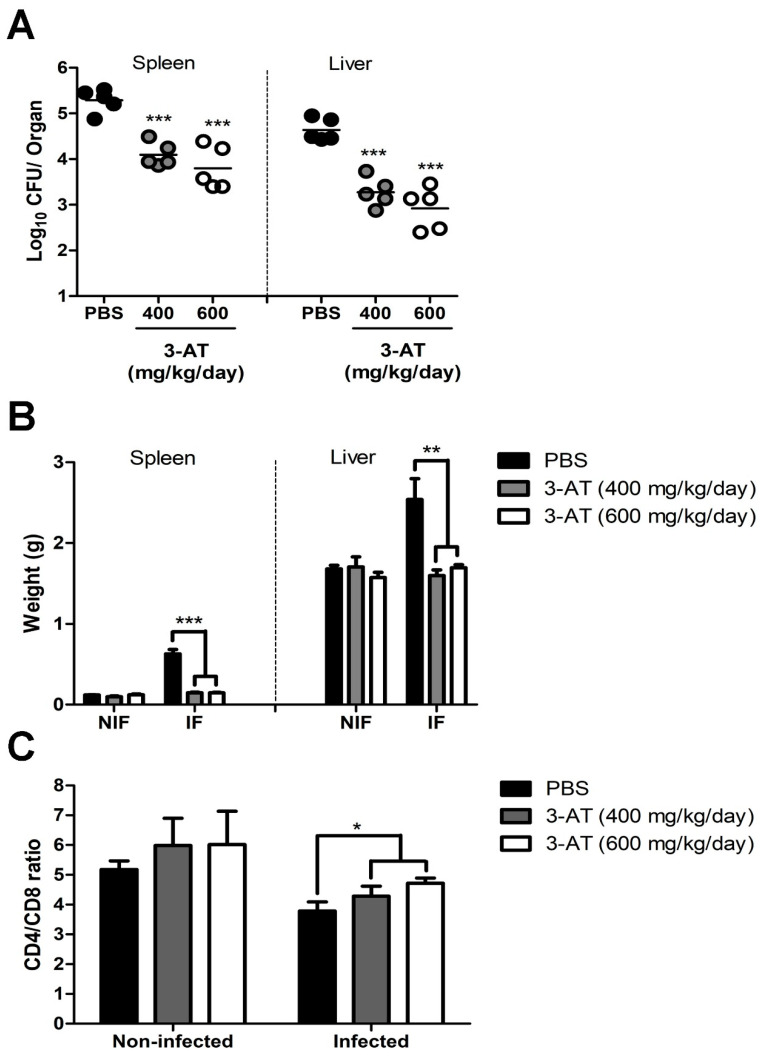
Protection against *B. abortus* in ICR mice treated with 3-AT. ICR mice were orally administered 400 or 600 mg/kg/day of 3-AT or a vehicle control three days prior to infection with *B. abortus.* The treatment was then intermittently continued for 12 days. At day 12 pi, the bacterial loads in the spleens and livers (**A**) and the total weights of the spleens and livers (**B**) were determined. The CD4^+^/CD8^+^ ratio in the blood of infected mice was analyzed via flow cytometry at day 12 pi (**C**). The data are presented as the mean ± SD of the mean of each group with five mice. Asterisks indicate statistically significant differences (* *p* < 0.05, ** *p* < 0.01, *** *p* < 0.001).

**Figure 6 ijms-24-17352-f006:**
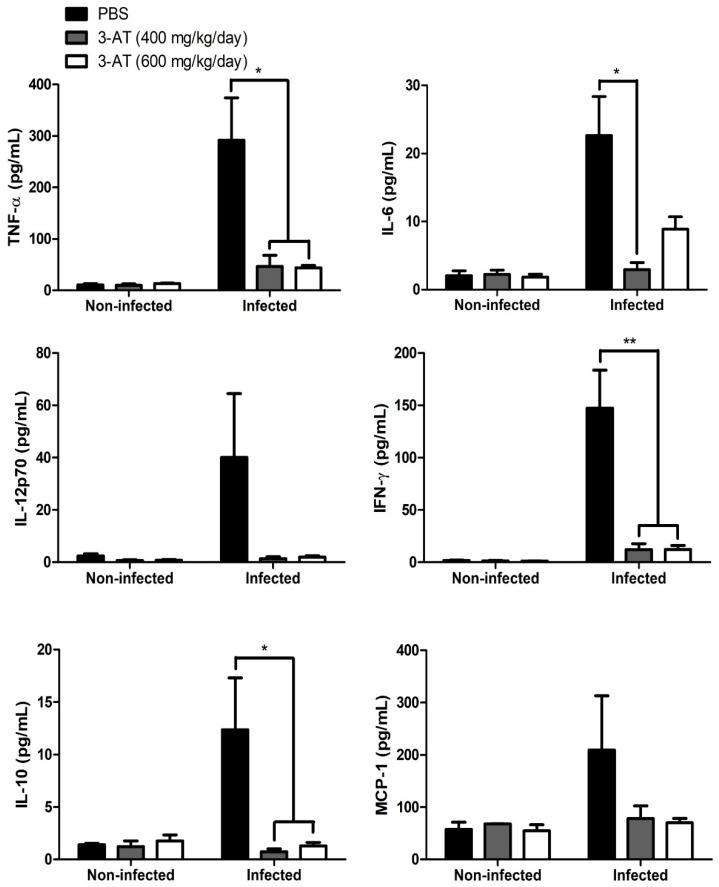
The compound 3-AT affects cytokine production in *B. abortus*-infected mice. The TNF-α, IL-6, IL-10, IFN-γ, IL-12p70, and MCP-1 levels in the sera from mice were measured using a CBA kit and flow cytometry at day 12 pi. Asterisks indicate statistically significant differences (* *p* < 0.05, ** *p* < 0.01).

**Table 1 ijms-24-17352-t001:** Mouse primer sequences used for qRT-PCR.

Gene Symbol	Gene Name	Primer Sequences
*GAPDH*	Glyceraldehyde-3-phosphate dehydrogenase	F: 5′-GGAGAAACCTGCCAAGTATG -3′ R: 5′-AACCTG GTCCTCAGTGTA-3′
*IL*-*6*	Interleukin 6	F: 5′-ACCACGGCCTTCCCTACTT-3′
		R: 5′-CATTTCCACGATTTCCCAGA-3′
*IL*-*10*	Interleukin 10	F: 5′-CATTTCCACGATTTCCCAGA-3′
		R: 5′-CATTTCCACGATTTCCCAGA-3′
*IL*-*12p40*	Interleukin-12 subunit p40	F: 5′-CTGGTGTCTCCACTCATGGC-3′
		R: 5′-GCGTGTCACAGGTGAGGTTC-3′
*TNF-α*	Tumor necrosis factor-alpha	F: 5′-CAGGTTCTGTCCCTTTCACTCACT-3′ R: 5′-GTTCAGTAGACAGAAGAGCGTGGT-3′

## Data Availability

Data are contained within the article.

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
