# Peer review of "Intracellular Growth Inhibition and Host Immune Modulation of 3-Amino-1,2,4-triazole in Murine Brucellosis"

_ijms, 2023, doi:10.3390/ijms242417352_

Round 1
Reviewer 1 Report
Comments and Suggestions for Authors
Dear Authors, this manuscript is important for the knowledge to supply in the livestock area and because it is an important zoonosis in Public Health. I am attaching some comment to have more clarity about your information.

Author Response
Decemmber, 2023
Dear Ms. Jocelyn Li
Section Managing Editor of Molecular Immunology section, International Journal of Molecular Sciences.
Enclosed hereby, please find our revised manuscript (ID: ijms-2720423) entitled: "Intracellular growth inhibition and host immune modulation of 3-amino-1,2,4-triazole in murine brucellosis".
We would like to thank the Editors and the reviewers for their constructive comments, which we considered when revising our manuscript. All of the comments raised by the reviewers have been addressed in detail in our efforts to improve our manuscript, and all the changes that we made in response to the reviewers' comments are highlighted in the yellow text in the revised manuscript. A point-by-point response to the reviewers' comments follows on the accompanying pages.
We hope that the revised manuscript is now acceptable for publication in International Journal of Molecular Sciences. Please contact me with any questions concerning the manuscript. I can be reached at +82-55-772-2359 or by e-mail at [email protected].
With regards,
Suk Kim, Ph. D.
College of Veterinary Medicine,
Gyeongsang National University,
Jinju, 660-701, Republic of Korea
Author's Responses to the Reviewer's Comments
ID: ijms-2720423
Title: Intracellular growth inhibition and host immune modulation of 3-amino-1,2,4-triazole in murine brucellosis
Authors:
Comments from the reviewers
# Reviewer 1
Q1: Line 17. I suggest looking for a synonymous of "inhibition" to avoid pleonasm
Authors answer to Q1:
As peer reviewer's suggestion, we replaced "inhibition" with "suppression" in line 17, abstract section.
Q2: Line 18. Was it a commercial inhibitor? Could author include this information?
Authors answer to Q2:
3-AT is a commercial product. However, the commercial 3-AT is used for purposes other than as a catalase inhibitor because commercial grade 3-AT generally contains catalase anti-inhibitory impurities. 3-AT, which is usually used for catalase inhibition studies, has higher purity and is used for study only. Therefore, we do not mention it in our manuscript.
Q3: Line 19. Before abbreviation, write full name
Authors answer to Q3:
The full name of Brucella abortus was written in line 16.
Q4: Line 53 and 56. Please, include a reference
Authors answer to Q4:
We added a reference for the sentence: "Furthermore, macrophages themselves rely on catalase to regulate oxidative compounds and safeguard themselves against oxidative stress and damage [8]." in line 53. Moreover, we added a reference for the sentence: "This heme enzyme converts H2O2 into water and oxygen, thereby reducing the toxic effects of H2O2 and protecting cells from excessive damage [9]." in line 56.
References:
- Tan, H. Y.; Wang, N.; Li, S.; Hong, M.; Wang, X.; Feng, Y., The Reactive oxygen species in macrophage polarization: reflecting its dual role in progression and treatment of human diseases. Oxid Med Cell Longev 2016, 2016, 2795090
- Röhrdanz, E.; Kahl, R., Alterations of antioxidant enzyme expression in response to hydrogen peroxide. Free Radic Biol Med 1998, 24, (1), 27-38.
Q5: Line 77-8. This kind of information could be included in the discussion.
Authors answer to Q5:
We moved the information: "The supplementation of catalase has been indicated to increase the intracellular growth of B. abortus within macrophages" in lines 77-78 following the reviewer's suggestion to line 202-203 in the discussion section following the reviewer's recommendation.
Q6: Line 99. DW =? Distillate water? Did you find cell viability? Perhaps I did not understand could you clarify this information?
Authors answer to Q6:
We corrected the abbreviation "DW" in Figure 1A to "PBS", and we also clarified that "DW" refers to distilled water in line 361 in the material and method section. The percentage of cell viability under 3-AT treatment, compared to control (PBS), is presented in Figure 1A. The primary objective of this experiment is to identify the non-cytotoxic concentrations of 3-AT for further studies. As depicted in Figure 1A, concentrations of 3 and 6mM of 3-AT show the highest cell survival, similar to the control. Higher concentrations result in a lower percentage of surviving cells, indicating cytotoxic effects caused by increased concentrations of 3-AT.
Q7: Line 133. NF-kB is a big family; perhaps there is another transcriptional factor involved. With this comment, before confirmation my suggestion is considered other studies in a near future
Authors answer to Q7:
Thank the reviewer for the insightful comment. We agree that NF-kB is a big family, and we appreciate your consideration of the possibility that other transcription factors may be involved in reducing the nucleus translocation of NF-kB. To address the reviewer's suggestion, we plan to conduct further studies to investigate the potential involvement of other transcription factors.
Q8: Lines 133-140. These results sound interesting. However, it is important to consider the IL10 as important gene regulator for many cytokines, but it did not show any response (RT or ELISA). The macrophages are the target of IL-10, for this reason did authors suggest the inhibition of the NF-kB transcript factor? I think that you need to carry out more studies; even other transcript factors might be included
Authors answer to Q8:
We appreciate the reviewers' attention to the role of IL-10 and the importance of your raised regarding its lack of response in regulating other cytokines both in qRT-PCR or in the cytokine level measured using CBA mouse inflammation kit and flow cytometry. While it is true that IL-10 is a crucial gene regulation of many cytokines and the macrophage is the target of the IL-10, the cytokine regulation in macrophage is complex. IL-10, while suppressing the production of pro-inflammation cytokines and chemokines, is also negatively regulated by TNF-α and IFN-β. Moreover, macrophages not only serve as the target of IL-10 but also as a source of IL-10 production. Since the regulation between the cytokines during infection is lacking in this study, it is necessary to conduct further study about the response of IL-10 as well as the other cytokines during 3-AT treatment during B. abortus infection.
The decision to focus on NF-kB translocation inhibition was based on a previous study by Jiang associates, which demonstrated that 3-AT affects the activation of NF-kB. This direction was chosen because NF-kB plays a vital role in regulating cytokine genes. However, as the reviewer rightly points out, other transcription factors might also be involved in cytokine regulation. In response to this suggestion, we plan to examine additional transcription factors to comprehensively elucidate the mechanism underlying cytokine regulation during Brucella infection and 3-AT treatment.
Q9: Line 184. The IL10 is a playtropic cytokine; it has important function, as regulator and contributes to reduce other cytokine activity. I am not quite sure of the IL10 function, perhaps there is another, or time to analyse experimental sample, check it please.
Authors answer to Q9:
We agree that IL-10 functions are diverse and complex, and there may be additional aspects to consider. We will carefully examine the experimental sample to ensure accurate measurements and explore any potential interaction or contribution of IL-10 in modulating other cytokine activities.
Q10: Line 279. Sorry, what did you mean with "increases the external H2O2 sensitivity of B. abortus?
Authors answer to Q10:
In this context, we want to convey that 3-AT treatment might reduce the catalase activity of intracellular B. abortus, leading to an increase in the sensitivity of this bacterium to the H2O2 within the host cell's intracellular environment. We used the term "external H2O2" to refer to the H2O2 produced by the host cell but not by Brucella itself.
Q11: Line 259. This reference is correct, other pathway mechanism might participate in the DNA-biding reduction for 3-AT, it is a possibility.
Authors answer to Q11:
Thank the reviewer for the comment regarding line 259 and the acknowledgment that other pathway mechanisms might participate in the DNA binding reduction for 3-AT.
Q12: Line 274. The lack of IL-10 might increase the pro-inflammatory cells, but this should be regulated, otherwise brucellosis might continue, without protection. Your results were relevant, but I suggest being careful with confirmed them as protective
Authors answer to Q12:
We acknowledge the reviewer's concern and agree that the balance between pro-inflammatory and anti-inflammatory responses in the context of infection, particularly in the context of Brucella infection. While our results indicate relevance, we understand that we must confirm it carefully. When we checked the cytokine level in cell culture media and in the serum of the infected mouse, we found that despite IL-10 reduction, the pro-inflammation cytokine such as TNF-α, IL-6 in 3-AT treated-cell culture media did not change or even lower than high control (Figure 4). In the in vivo experiment, the level of pro-inflammation cytokines in the blood of 3-AT-administrated mice was lower than in the control group, although IL-10 in this group was lower than in the control group (Figure 6). These results indicate that 3-AT treatment reduces IL-10 production but does not lead to excess pro-inflammatory cytokine, suggesting the regulation of cytokine under 3-AT. However, we agree with the reviewer's suggestion of being careful with the conclusion protective effect of 3-AT. Therefore, we replaced the "protective effect" by "potential effect" to change infection in line 158.
Q12: Line 280. Did you mean the ELISA assay results?
Authors answer to Q12:
We did not use the ELISA assay to measure the cytokine level. As described in the material and method, the cytokine levels were measured using a CBA mouse inflammation kit, following manual instructions. Data were acquired and analyzed using BD FACVerse flow cytometer and FCAP array software.
Q13: Line 318. Considering that 3-AT is a synthetic reactive, I wonder if author have thought how will be the evaluation in domestic ruminants, because the response of small mammals could be different to the definitive host.
Authors answer to Q13:
We agree with the reviewer that mice are not natural hosts of Brucella, and these small animals might have different responses to infection or treatment. Typical natural hosts of Brucella are sheep, cattle, pigs, and humans. Researching the pathophysiology of brucellosis in natural hosts would be ideal, but it's not always practical or ethical. As a result, researchers often use small laboratory animals like mice as models instead. While the results obtained from such models may not be immediately applicable to humans or the target species, they still provide valuable insights. With the advent of inbred, mutant, knockout, and transgenic mice, in addition to a better understanding of their biology and immunology, the mouse has become the standard model for brucellosis research. Given our lab's conditions, conducting experiments on mice is the most appropriate option available. Researching the pathophysiology of brucellosis in natural hosts would be ideal, but it's not always practical or ethical. As a result, researchers often use small laboratory animals like mice as models instead. While the results obtained from such models may not be immediately applicable to humans or the target species, they still provide valuable insights. With the advent of inbred, mutant, knockout, and transgenic mice, in addition to a better understanding of their biology and immunology, the mouse has become the standard model for brucellosis research. Given our lab's conditions, conducting experiments on mice is the most appropriate option available.
Q14: Material section: I did not found references in this section; I think that authors have established their own methodology. I suggest include a second housekeeping in a future to normalized data. Did authors design your own primers (sequence?). Which program did author used to analyze the expression level?(GraphPad?)
Authors answer to Q13:
Thank the reviewer for concern about the references in the material and method section. However, most of the assays we used in this study followed the guidance in the kit or chemical we bought for assay, the other method we established by ourselves. We performed a catalase activity assay in Brucella following the previous study. The reference for this method was cited in line 370.
We appreciate the suggestion of using the second house keeping gene for normalized results of real-time PCR rather than just using GAPDH. Using the second house keeping gene for normalization might improve real-time PCR's accuracy, reliability, and reproducibility. Therefore, we are going to use the second reference gene in the next real-time PCR experiment.
We mentioned Bio-Rad CFX Maestro software as the program that we to analyze the expression level of mRNA in line 426.
# Reviewer 2
Nguyen et al. demonstrate in the presented manuscript the protective effects of the catalase inhibitor 3-AT against B. abortus. The authors present interesting results that show that inhibiting catalase activity could be a novel approach in brucellosis. Few points should be considered in a revised version of the manuscript.
Q1: The authors present data on the toxicity of 3-AT on RAW cells. It is not clear whether they determined the combined cell death of 3-AT and infection with B. abortus.
Authors answer to Q1:
We appreciate the valuable comment about assessing the combined effect of 3-AT and B. abortus infection on RAW cells. In our study, we focused primarily on the individual toxicity effect of 3-AT on RAW cells to establish the baseline understanding of 3-AT effect in isolation before asses the interaction with Brucella infection. However, based on the reviewer's comment, we realize that it is essential to examine the cell death, including necrosis, apoptosis, and autophagy in the infected RAW cells in the presence of 3-AT to give a comprehensive view of the mechanism of reducing intracellular Brucella within RAW cells.
Q2: The authors show via western blot the translocation of NFkB into the nucleus. The western blot is missing a control that shows the purity of the nuclear protein extracts. Cytosolic protein levels should be included in the blot.
Authors answer to Q2:
We acknowledge the importance of the control demonstrating the purity of nuclear protein extracts in the western blot analysis, particularly by including cytosolic protein levels. When we were conducting nucleus protein extraction, to confirm the purity of the nucleus protein, we only checked the protein expression level of Lamin B1, which is usually used as a control nucleus protein in both the nucleus and cytosolic extraction. We observed vigorous intensity of Lamin B1 at size 65 kDa in nucleus extraction. However, only the weak intensity of additional Lamin B1 45 kD was observed in the cytosol, but not Lamin B1 65 kD. That demonstrates the purity of nucleus protein. We checked the lamin B1 expression in the cytosol of some samples just to confirm the effectiveness of the nucleus protein extraction method Therefore, due to limitations in the experimental design, we excluded these results from the figure. Our initial purpose was to focus on investigating the NF-kB expression within the nucleus. Hence, the level of NF-kB in the cytosol was not accessed. We will address this shortcoming in the next study.
Q3: The nuclear extracts show strong bands of p105 and p100. Canonically cleavage is required for translocation into the nucleus.
Authors answer to Q3:
We agree with the reviewer's comment that canonical cleavage is required for translocation into the nucleus. P105 and P100 are precursors of NF-Kb p50 and p52 and are primarily localized in the cytosol in their unprocessed forms. In addition to the canonical NF-kB activation pathway, a non-canonical pathway can directly process p100 into the nucleus. Figure 3 shows that p105 is expressed strongly only in Brucella-infected cells, which may be due to the cellular stress triggered by the infection at 48 hours. The cellular stress and infection can activate alternative NF-kB, cytokine secretion, ROS accumulation, and other signaling pathways, leading to NF-kB p100 and NF-kB p105 in the nucleus.

Reviewer 2 Report
Comments and Suggestions for Authors
Nguyen et al. demonstrate in the presented manuscript protective effects of the catalase inhibitor 3-AT against B. abortus. The authors present interesting results that show that inhibiting catalase activity could be a novel approach in brucellosis. Few points should be considered in a revised version of the manuscript.
The authors present data on the toxicity of 3-AT on RAW cells. It is not clear whether they determined the combined cell death of 3-AT and infection with B. abortus.
The authors show via western blot the translocation of NFkB into the nucleus. The western blot is missing a control that shows the purity of the nuclear protein extracts. Cytosolic protein levels should be included in the blot.
The nuclear extracts show strong bands of p105 and p100. Canonically cleavage is required for translocation into the nucleus.
Author Response

(The authors gave the same response as above.)
